# Sustainable Production Policy Impact on Palm Oil Firms' Performance: Empirical Analysis from Indonesia

**Noxolo Kunene and Yessica C.Y. Chung \***

Department of Agribusiness management, National Pingtung University of Science and Technology, Pingtung 912, Taiwan; kunenenoxolo01@gmail.com
**\*** Correspondence: yessicachung@mail.npust.edu.tw

**Abstract:** Sustainable production is a key element of sustainable development. The concept was first introduced in the United Nations Rio Earth Summit in 1992 and has become an important item on the management of industries. In conjunction, the government of Indonesia introduced the Indonesian Sustainable Palm Oil (ISPO) policy in 2011 to adhere to international sustainability standards of Sustainable Palm Oil and of reducing emissions from deforestation and forest degradation (REDD+). This study investigates the impact of ISPO policy on palm oil firms' performance. Using a sample of 409 palm oil firms of the Indonesian palm oil sector for the years 2010 and 2015, we employed a regression discontinuity (RD) with a difference-in-differences approach to explore the effect of the policy on firms' performance. The RD results show that the introduction of the policy significantly reduced large firms' profit by IDR 75m (equivalent to USD 5250); the negative effect of the policy increased with firm size. Furthermore, there was a significant reduction in performance for firms that promptly purchased land before the policy's ban on land expansion. These findings suggest that a punitive sustainable production policy does not sustain the palm oil sector. Nevertheless, large companies that complied with international sustainability measures ahead of the introduction of the domestic policy benefited.

**Keywords:** sustainability production policy; palm oil; firm performance; regression discontinuity design; Indonesia

## 1. Introduction

Sustainable production presents a profound channel for achieving sustainable development. Since the highlight of the concept in the Rio Earth Summit in 1992, sustainable production has become important. More specifically, a total of 108 countries had put in place national policies related to sustainable production and consumption in 2018 [1]. According to the United Nations [1], sustainable production is the approach of minimising environmental depletion from production systems while meeting the needs of all. Driven by different factors, there is a growing number of industries underpinning the sustainable production [2]. Previous studies revealed that sustainable productions increase sales [3], competitive advantage [4], operational efficiency [5], and profit margins [6]. These studies suggest that sustainable production systems improve firms' performance. Despite these findings, the debate about sustainable production and firm performance still dominates. This debate is because of the various driving factors of sustainable production.

A large body of literature documented driving factors of implementing sustainable production. A study by Price et al. [7] examined the link between a sustainability policy, company characteristics, and the application of sustainable business practices. The study found that large companies implement

sustainable business practices. Meanwhile, Schrettle et al. [8] identified the application of modern manufacturing technologies, greener products [4], and green practices as the prominent pathways of dealing with sustainability successfully. Though, for companies to foresee sustainability and financial gains, they must strategically organise unique resources and capabilities [6]. Zhu [9] observed that market and public pressures drive companies to implement sustainable production with the exception of land-saving practices, whereas government regulations encourage companies to adopt sustainable practices related to water, land, and energy. However, companies require support for land and energy-saving practices. Overall, these studies suggest that sustainable production is driven by market pressures, technological factors, public interest and government regulations. Efforts by the government are motivated by international community policies to implement sustainable production at the local level. The translation of international sustainable production policies at the national level is different with each nation. The government can adhere to international policies by supporting sustainable production systems, or by enforcing strict regulations. The literature on the impact of sustainable production driven by the latter is limited.

This narrowed focus prompts us to address this need, using the case of the Indonesian palm oil industry. We empirically examine the impact of a sustainable production policy on firms' performance. In March 2011, the government of Indonesia introduced the Indonesian Sustainable Palm Oil (ISPO) policy, which is a mandatory standard for all large palm oil producers in the country. We use the case of Indonesia because of the following reasons. First, palm oil has become the most important commodity in the vegetable oil industry. Hence, Indonesia is the world-leading producer of palm oil, producing more than half of the total palm oil products in the world, and contributing to 3.5% of its GDP [10]. Second, the formulation of ISPO marks a historic milestone for Indonesia as the first southern country to respond to the negative campaigns and preeminence of international sustainability standards that present the values of western countries [11–13]. More specifically, ISPO has deemed to support the REDD+ policy and abide by the Amsterdam Declaration of a Fully Sustainable Palm Oil Supply Chain by the end of 2020 [14]. Third, from the context of Indonesia, scholars have paid close attention to the environmental impacts of the industry such as carbon saving, palm oil-based-biofuel energy, and the effect of REDD+ projects. However, none of these studies advances our knowledge of the impact ISPO on firm performance. A comprehensive examination of this impact is critical for guiding policymakers [15].

We seek to bridge this gap by empirically investigating the impact of ISPO policy on palm oil firms' performance. Using census surveys of Indonesian manufacturing industry for the years 2010 and 2015, we employ a regression discontinuity approach to estimate the impact of ISPO policy on firms' performance. The design of the policy allowed us to explore the discontinuity in firms' performance as the principles of the policy differ according to firm size. Furthermore, other factors may affect firms' performance. Therefore, we use the difference-in-differences (DID) model to show the effect of these factors. Our outcome measure is net profit. It has been shown that net profit is an important measure of firms' financial performance [16].

Explicitly, we aim to: (1) investigate the causal effect of ISPO policy on palm oil firms' performance and infer the causal effect of firm and provincial characteristics, and geographical features; (2) determine the specific type of firms that are affected by the policy. The contributions of this study are twofold. First, this study is among the few studies that empirically explore the impact of a policy-driven sustainable production on firm performance. Prior studies may have been constrained by the availability of data, which is scarce. Yet, we have access to a rich database that contains information about manufacturing industries in Indonesia. Second, to the best of our knowledge, it is the first study that employed a novel quasi-experimental approach, called regression discontinuity (RD) to empirically identify the effect of sustainable production policy on firm performance. This holistic approach allowed us to estimate the causal effect of a randomly selected sample of firms with similar nature, thus addressing the issue of bias such as self-selection bias.

The rest of this study is structured as follows. Section 2 provides background on the Indonesian palm oil industry and outlines the ISPO policy context. Section 3 describes data and presents details on empirical models. Section 4 presents the empirical results. Finally, the last section presents discussions and policy implications.

## 2. Indonesian Palm Oil Industry

Indonesia became the largest palm oil producer in 2007, surpassing Malaysia producing 62% of all palm oil in the world in 2018 [17]. Since then, the palm oil industry in the country has been developing rapidly. Along with the development of plantation areas, the volume of production increased significantly by one million hectares in 2000 [18]. The industry plays a predominant role in supporting economic development in Indonesia, contributing to 3.5% of GDP in 2019 and employing 4.2 million people in 2019 [19]. Meanwhile, from the perspective of consumption, palm oil is an imperative commodity because it is used as the main raw material for edible oil and in non-food products such as detergents, industrial inputs and biofuel. In terms of geography, Riau Province (in Sumatra Island) is the leading palm oil origin in the country, followed by North Sumatra, Central Kalimantan, South Sumatra, and West Kalimantan. Most of the Indonesian palm oil is tailored for export and the most important export destinations are China, India, Pakistan, Malaysia, and the Netherlands. Thus, palm oil exports have become an imperative foreign exchange earner for the country, with a total value of USD 21.4 billion export in 2018 [19].

Recent events beget doubt over the status of Indonesia as the most significant producer of palm oil. Specifically, the recent incidence of fires in Sumatra and Kalimantan rejuvenated excessive media attention and reputation of environmental damage. This incidence stems from the traditional technique of land clearing for oil palm cultivation. Furthermore, the production of palm oil in Indonesia is associated with environmental depletion, such as an 83% decline in forestry butterfly species and 79% of forest conversion [20]. Allied to the public scrutiny, the issue of reducing emissions from deforestation and degradation (REDD+) dominated, pressuring the Indonesian government and palm oil producers to take action. REDD+ is a climate change mitigation mechanism of reducing emissions in developing countries. The government of Indonesia acted in response, by introducing the Indonesian Sustainable Palm Oil policy, known as ISPO in March 2011. This policy is aimed at ensuring growers' compliance with land use regulations and human rights, support REDD+ commitments, and conserve the environment, thereby improving the sustainability of the industry and the competitiveness of the industry in the global market [19]. Principally, ISPO is a mandatory standard for large firms in the palm oil sector that is economically, socially, and environmentally viable based on existing law in Indonesia [19] (efforts of enforcing ISPO certification for small-medium firms are underway [14]). Thus, large firms are certified in 2015, whereas medium firms are not ISPO certified as the policy does not apply to them [21]. The Indonesian central bureau of statistics defines medium firms as those with the employment of 5–99 people and large firms as those with 100 and or more employees.

ISPO is defined under seven principles. These criteria are devised to curtail the killing of endangered species, greenhouse gases, child labour, and deforestation. Before attaining an ISPO certificate, companies must follow a pre-set of procedures. The procedures start with the assessment by the provincial government [22] under which companies are assessed on their compliance with national regulations and plantation management or supply base. Only companies that score well in this stage proceed to the next stage. Unsuccessful companies are allowed to adjust their practices and undergo the first evaluation. Following a positive evaluation by the ISPO assessment team, companies are granted an ISPO valid certificate for 5 years. Large companies that failed to apply for ISPO certification face a sanction mechanism, which is the revocation of business permits license [22].

## 3. Data and Methods

### 3.1. Data

We constructed a pooled database of medium and large palm oil processing firms in Indonesia for the years 2010 and 2015. These data were obtained from the Indonesian manufacturing industry census surveys conducted by the Indonesian central statistics agency (BPS-Statistics Indonesia). This census survey collects annual information about medium and large-scale manufacturing industries on income, expenditure, production, capital, and location. More specifically, the information on the survey specifies the characteristics of the firms. The first type of information on the survey was the firm identification number, location, and ownership structure of the firms. The firms' identification number, allowed us to acquire a balanced data of two-years (2010 and 2015). The rest of the information relates to the value of expenses, production (goods produced), and capital, respectively. The final sample consisted of 406 firms. We further supplemented the census data with provincial and regional data.

According to Uning et al. [23], local and regional changes such as climatic conditions and land-use may impose significant changes on the palm oil industry; therefore, there is a need to assess these impacts. The provincial data detailing the provincial and regional characteristics were obtained from the Indonesian Bureau of Statistics (BPS). All provincial and regional data are based on 2010 and 2015 data, with the exception of education data. The census survey and regional data allowed us to construct a set of variables for our analysis. These constructed variables highlighting firm and provincial characteristics are presented in Table 1. Empirical evidence has shown that firm characteristics such as capital structures [24], firm size, energy efficiency [25] and clustering of firms [26] affect firms' financial performance. Scholarship on the effect of capital structures and firm performance has shown that foreign capital positively influences firm performance due to high commitment and larger shareholding [27]. By the same token, domestically acquired firms in the United States performed poorer than firms acquired by foreign investors in terms of labour productivity [28]. In contrast, the relationship between firm size and performance has received the highest attention in strategic management research. Studies have shown that firm size positively influences firm performance due to their capability to withstand different dynamics [29] and cost advantage presented by economies of scale [30].

**Table 1.** Description of variables.

| Variable | Definition | Units |
| --- | --- | --- |
| | **Dependent variable** | |
| Performance | Value of net profit | IDR millions |
| | **Independent variable** | |
| | *Firms characteristics* | |
| N. of employees | Number of employees in a firm | - |
| Medium firms | If number of employees $20 \geq 99$ | - |
| Large firms | If number of employees $100 \geq$ | - |
| Domestic capital | Percentage of capital owned by Indonesian private individuals | % |
| Foreign capital | Percentage of capital owned by Foreign investors | % |
| Location | If located in the industrial area (=1) | - |
| Technology | Quantity of electricity generated from palm oil mill effluent | million / Kwh |
| Energy efficiency | Ratio of total electricity use to operating revenues | kwh/IDR |
| Land expansion | Value of land purchased by firms | IDR thousand |
| Export | If firm export products (=1) | - |

**Table 1.** *Cont.*

| Variable | Definition | Units |
|---|---|---|
| | *Provincial characteristics* | |
| Population | Total number of people | thousands |
| Rainfall | Quantity of annual rainfall | million mm |
| Temperature | Average annual temperature | °C |
| Education | Number of higher-educational institutions | |
| Water supply | Quantity of cleaned water distributed | million m$^3$ |
| Agricultural land | Total area of arable land | million ha |
| REDD+ | If located in a province that implemented REDD+ program (=1) | |
| | *Geographical region* | |
| Java | If located in Java island (=1). | |
| Papua | If located in Papua province (=2). | |
| Sulawesi | If located in Sulawesi island (=3). | |
| Kalimantan | If located in Kalimantan island (=4). | |
| Sumatra | If located in Sumatera island (=5). | |

On one hand, the issue of environmental depletion forced firms to do more than just preventing pollution but to reduce energy consumption. It is considered that the adoption of energy-efficient technologies indirectly explains the variation of economic performance such as cost-saving [31]. Meanwhile, a large body of literature revealed that the adoption of technology improves operational efficiency, thus enhancing the competitiveness of agribusiness and their access to international markets [32,33]. Because of the association of these firm characteristics with firm performance, we included them in our analysis as control variables.

Klassen and McLaughlin [34] showed that financial performance is one measure of identifying firms' benefits from embedding sustainable production. Similar to previous studies [35,36], we use net profit as the proxy for financial performance because of the following reason. First, net profit is the most comprehensive measure of companies' profitability [17]. Second, one of the principal objectives of ISPO is to enhance palm oil firms' economic benefits through improved operational efficiency. We define net profit (net income) as the difference between operating profit and the total cost of production. Moreover, to avoid attributing the impact of REDD+ to ISPO policy, we included a control variable (dummy) to capture the potential impact of the distinct policy intervention. In addition, we included a dummy variable measuring the impact of location factor that is geographic location, which indicates the total area of oil palm plantation in a region. We consider the region with the least oil palm plantation as the baseline (Java = 1) and, likewise, a region with a large area of oil palms as the strong supply base (Sumatra = 5).

*3.2. Empirical Framework*

Our main empirical framework is to use a regression discontinuity (RD) approach [37] to estimate the treatment effect of ISPO policy on the treated firm performance. The fundamental concept of RD design is to evaluate the causal effect of a program, using a forcing variable, also called an assignment variable. This observable variable has a fixed threshold value, called the cut-off point, which determines treatment status. Specifically, RD compares the treatment group with x values just above the cutoff and control group with x values just below the cutoff, assuming that the variation in the treatment near the cutoff is approximately randomised [38]. The RD approach can be sharp or fuzzy. Sharp RD is the case where the assignment is deterministic, whereas fuzzy RD occurs when "imperfect" implementation by program participants affects the probability of program participation. Imperfect implementation occurs when some individuals who are illegible to the program are treated due to spill-over effects. We use a sharp RD framework in this study since the treatment is solely determined by firm size at the cut-off value. Our assignment variable is firm size. Since large firms are ISPO certified (treatment group) and medium firms are uncertified (control group). Our cut-off value is 100.

One distinct feature of the RD approach is that as firm size approaches the cut-off value from above and below, the firms in both groups (the control and treatment group) become more similar on both observable and unobservable characteristics [39]. These conditions suggest that these two firm groups are comparable in this neighbourhood around the cut-off value. It is this sample of firms that are under investigation. This assumption is difficult to validate in other commonly used quasi-experiments, such as Propensity Score Matching.

The major assumption in the RD framework is that the performance of firms below and above the cut-off value is continuous in the absence of ISPO policy and firms become ISPO certified based on firm size. Let $Y_i(0)$ and $Y_i(1)$ present the potential outcomes for firm $i$ and $S$ *to* present firm size, where $Y_i(0)$ is the performance of uncertified firms and $Y_i(1)$ is the performance of firms that are ISPO certified. Let $Z_i \in \{0, 1\}$ present treatment; $Z_i$ equals one if firm $i$ is ISPO certified, and zero otherwise. Therefore, the observed outcome can be outlined as follows:

$$Y_i = (1 - Z_i)Y_i(0) + Z_iY_i(1) = \begin{cases} Y_i(0), & if \ Z_{i(S_i)} = 0 \\ Y_i(1), & if \ Z_{i(S_i)} = 1 \end{cases} \tag{1}$$

We examine the treatment effect near the cut-off as $S$ approaches the cut-off C:

$$TE_{SRD} = \lim_{s \uparrow c} E\lfloor Y_i(1)|S_i = s\rfloor - \lim_{s \downarrow c} E\lfloor Y_i(0)|S_i = s\rfloor \tag{2}$$

Based on the aforementioned assumptions, the performance of firms will be continuous at the cut-off in the absence of ISPO policy. The continuity assumption will be violated if firms could manipulate their size at the cut-off. An example of manipulation is when firms suddenly scale down to exclude themselves from ISPO certification. To confirm that the discontinuity observed is due to ISPO introduction and is not confounded by manipulation, regression models are used to test for the smooth function [40]:

$$Y_i = \beta_0 + \beta_1 Z_i + \beta_2(Z_i - x) \ \beta_3 Z_i (Z_i - x) + \varepsilon_i \tag{3}$$

$Z = 1$ when a firm is ISPO certified and $\beta_1$ is the jump at the cut-off. $\beta_2$ is the slope in the absence of ISPO policy, and the interaction term's coefficient. $\beta_3$ allows for a different relationship between Z and Y on the other side of c.

Other factors may affect the performance of firms. We complementary employed a DID model to investigate the causal effect of firm characteristics, provincial and regional characteristics on firms' performance. The DID model is time-invariant; the model strips out any unobserved confounders between the treatment and control group that are fixed over time with the exception of those that coincide with the ISPO policy. We used a fixed effect to control for differences in characteristics between control and treatment group. Under the basic DID approach, the causal effect of the policy is modelled as follows:

$$Y_{it} = \beta_0 + \beta_1 ISPO_{it} + \beta_2 Large \ firm_{it} + \beta_3 ISPO \times Large \ firm_i + \varepsilon_{it} \tag{4}$$

where $Y_{it}$ denotes the performance of the $i$-th firm in year t; $Period_{it}$ is the policy dummy variable for the $i$-th firm, and it is equal to zero for year 2010 (pre-ISPO) and one for year 2015 (post-ISPO); $Large \ firm_{it}$ takes 1 if the $i$-th firm is large and 0 otherwise; $\beta_3$ is the most important coefficient to capture the ISPO effect on the treated (large firms); and $\varepsilon_{it}$ is an error term. To evaluate the causal effect of the aforementioned factors, we introduced a vector of covariates X. Thus, Equation (4) is outlined as follows:

$$Y_{it} = \beta_0 + \beta_1 ISPO_{it} + \beta_2 Large \ firm_{it} + \beta_3 ISPO \times Size_i + \sum_{n=1}^{19} \pi X_{it} + \varepsilon_{it} \tag{5}$$

where $X_{it}$ is the vector of covariates defined in Table 1, and $\pi$ is the vector of coefficients.

## 4. Results

Table 2 presents the performance of firms. Figures in this table compare the performance of medium and large firms over the 2010 and 2015 period. The performance of large firms decreased by IDR 377.93 m (222%) during the period of 2010–2015, compared to the decrease of IDR 143.79 m (230%) for medium firms in the same period. More specifically, the difference in performance between large and medium firms in 2010 was IDR 108.79 m, which is much higher than the difference of −125.35m in 2015. Meanwhile, results from a *t*-test analysis detected a 10% ($p = 0.08$) significant difference in performance in 2015 between the two groups, whereas no significant differences were observed in 2010. These results are in line with the key assumption (common trend assumption) of the DID model. This means that the performance of medium (control) and large firms (treatment) followed the same trend before the introduction of ISPO.

**Table 2.** Performance (Net profit in IDR million) of palm oil firms in 2010 and 2015.

| | All $n = 406$ | | Medium $n = 33$ | | Large $n = 373$ | | Difference Large-Medium | |
|---|---|---|---|---|---|---|---|---|
| | 2010 | 2015 | 2010 | 2015 | 2010 | 2015 | 2010 | 2015 |
| **Mean** | 162 | −196.90 | 62.04 | −81.75 | 170.83 | −207.10 | 108.79 | −125.35 * |
| | (473.1) | (395) | (−81.86) | (115.362) | (492.08) | (409.17) | | |
| **Mean change between 2010 and 2015** | −358.90 | | −143.79 | | −377.93 | | −234.14 | |
| | (471.10) | | (122.88) | | (490.17) | | | |

**Note**: Standard deviations are in the parenthesis. Figures below the mean change present the average profit for both years. Significance difference at * $p < 0.1$.

Overall, the results show a decrease in performance in both the treatment and control groups in 2015. The large drop in performance could be attributed to other factors, perhaps the severe drought in 2015 [40,41], coupled with a decline in global palm oil prices [24,42,43]. We dropped global price variable in our analysis as it was negatively and significantly correlated with the dummy variable period (year), indicating a descending trend in the global palm oil prices. Nonetheless, the effect of these two shocks on large firms was weaker than that of ISPO as the policy directly and precisely targeted large firms while natural and global events affected both medium and large firms. Thus, the effect of these naturally occurring factors is likely to be absorbed by the trend in the performance of both medium and large firms.

Descriptive statistics for firm characteristics, provincial characteristics, and geographical location for 2010 and 2015 are displayed in Table 3. Concerning ownership structure, there is a significant difference in the percentage of capital investment owned by Indonesian investors between the two groups; medium firms had a higher share of domestic capital compared with large firms. Moreover, we detected a significant difference in location between the two groups. On average, 27% of the large firms are located in the industrial zone, compared to 17% of their counterpart. Additionally, the ratio of energy efficiency is slightly lower for large firms compared to medium firms, indicating that large firms use electricity more efficiently compared to medium firms. These results indicate that medium firms lack skills and the ability to adopt energy-efficient measures.

**Table 3.** Descriptive statistics of selected variables.

| Variable | All | | Medium | | Large | | Mean Difference between Medium and Large (Large-Medium) | |
|---|---|---|---|---|---|---|---|---|
| | Mean | SD | Mean | SD | Mean | SD | | |
| *Firms characteristics* | | | | | | | | |
| N. of employees | 217.56 | 375.76 | 87.77 | 120.73 | 229.05 | 388.34 | 141.28 | *** |
| Domestic capital | 68.88 | 44.53 | 83.70 | 36.79 | 67.57 | 44.94 | −16.13 | *** |
| Foreign capital | 18.95 | 36.84 | 12.96 | 32.96 | 19.48 | 37.14 | 6.52 | |
| Location | 0.26 | 0.44 | 0.17 | 0.38 | 0.27 | 0.44 | 0.10 | ** |
| Technology | 1.11 | 5.44 | 0.85 | 3.87 | 1.13 | 5.56 | 0.28 | |
| Energy efficiency | 0.00 | 0.00 | 0.01 | 0.02 | 0.00 | 0.01 | −0.01 | *** |
| Land expansion | 781.01 | 5695.00 | 206.21 | 1477.90 | 832.00 | 5923.76 | 625.79 | |
| Export | 0.11 | 0.31 | 0.17 | 0.38 | 0.10 | 0.30 | −0.06 | |
| *Provincial characteristics and macro factors* | | | | | | | | |
| Population | 8197.18 | 5982.37 | 10,026.80 | 7659.27 | 8035.31 | 5789.49 | −1991.49 | *** |
| Rainfall | 3.06 | 5.62 | 2.03 | 1.76 | 3.15 | 5.83 | 1.13 | |
| Temperature | 22.36 | 10.33 | 21.09 | 11.54 | 22.47 | 10.22 | 1.38 | |
| Education | 130.89 | 128.33 | 200.15 | 227.66 | 124.76 | 113.77 | −75.39 | *** |
| Water supply | 99,972.25 | 108,887.80 | 133,025.50 | 124,716.30 | 97,047.97 | 106,979.00 | −35,977.53 | ** |
| Agricultural land | 1,393,866 | 551,102.10 | 1,249,167 | 473,780.50 | 1,406,668 | 555,892.20 | 157,501 | ** |
| REDD+ | 0.59 | 0.49 | 0.42 | 0.50 | 0.60 | 0.49 | 0.18 | ** |
| *Geographical region* | | | | | | | | |
| Java | 0.02 | 0.13 | 0.09 | 0.29 | 0.01 | 0.10 | −0.08 | *** |
| Papua | 0.01 | 0.11 | - | - | 0.01 | 0.12 | | |
| Sulawesi | 0.03 | 0.18 | 0.06 | 0.24 | 0.03 | 0.17 | −0.03 | |
| Kalimantan | 0.15 | 0.35 | 0.03 | 0.17 | 0.16 | 0.36 | 0.13 | *** |
| Sumatra | 0.79 | 0.41 | 0.82 | 0.39 | 0.79 | 0.41 | −0.03 | |
| **N** | 406 | | 23 | | 373 | | | |

Note: Significance difference at *** $p < 0.01$, ** $p < 0.05$.

Regarding provincial characteristics, demographics (population and education) and water supply are significantly higher in provinces occupied by medium firms. On the other hand, we observed that about 60% of large firms are located in provinces that implemented REDD+, which is significantly higher than medium firms. This could be attributed to the on-going REDD+ projects, which have been designated in areas with large oil palms plantations. Additionally, we observed a significant difference in agricultural land at the provincial level between the two groups. As expected, large firms are located in provinces with abundant agricultural resources. Furthermore, concerning geographical features, 82% of the medium firms are located in Sumatra Island compared with 79% of large firms. The potential reason is that the island has good infrastructure and services that are suited for the operation of palm oil firms

### 4.1. RD Analysis

Because the ISPO policy applies to large-scale firms, not to the medium firms, this scenario presents a cut-off value, which allows for the identification of ISPO effect by comparing the performance of the two groups. For the aforementioned reasons, we now branch to employ RD approach and later apply the DID model to investigate the impact of other factors on firm performance. We applied a local polynomial method with $p = 1$ to appropriately analyse the RD effect near the cut-off value. Table 4 presents the results. The coefficient of the treatment effect in 2010 is insignificant. This justifies the assumption that firms in 2010 are similar near the cut-off in the absence of ISPO policy. The treatment estimate is 79.03 m and significant at 5%. This result indicates that the ISPO policy reduced the performance of the treated firms (large firms) by IDR 79 m, which equals USD 5530 (As of December 2015, the exchange rate between IDR and USD is 0.00007 IDR/USD) and accounting for 46% of the average performance of large firms in 2010.

**Table 4.** RD estimate effect of Indonesian Sustainable Palm Oil (ISPO) policy on palm oil firms' performance.

|  | Coef. |  | S.E |
| --- | --- | --- | --- |
| Treatment dummy (pre-policy) | −69.774 |  | 48.991 |
| Treatment dummy (post-policy) | −79.030 | ** | 30.861 |
| **N** |  | 406 |  |

** $p < 0.05$. S.E is adjusted with cluster.

We re-examined the density of the forcing variable [44] by applying a local polynomial density estimator. The estimated difference in the density of the forcing variable at the cutoff is 1.513 with a *p*-value of 0.13 (see Figure A1). These insignificant results indicate that the density of firm size does not change sharply at the cutoff. Thus, firms do not manipulate their size to gain an economic benefit or to avoid complying with the policy. In furtherance, we tested covariate balance by performing robust local-polynomial inference to attain confidence intervals and *p*-values for effects on six predetermined covariates: technology, energy intensity, rainfall, temperatures, Java, and Sumatra (see Figure A2). The results show no notable or significant covariates difference at the cut-off between medium and large firms. This is encouraging and justifying the adequacy of using firm size as the forcing variable. This implies that results in this study are consistent with the cogency of the RD design. This local average treatment effect at the cut-off point is highlighted graphically using RD plots [45] (see Figure A3).

### 4.2. DID Analysis

We further examine the ISPO effect by using DID to control the effect of firm characteristics, province characteristics, and location factors. Table 5 shows the results for four specifications that examine the ISPO effect in different angles. The first specification is constructed upon the selection of the window around the cutoff (that is, within 0.5 points bandwidth) in RD analysis. Within the bandwidth, there are 158 firms, in which 24 are medium firms and 134 are large firms. In the second

model, we used a full sample and in the third model, we applied a continuous variable that is the number of employees as an alternative of the dummy variable to capture the marginal effect of firm size. Lastly, we added an interaction term of ISPO. We used fixed effect in all model specifications to control for changes over time.

We are interested in the coefficient of the interaction term of ISPO implementation and large firms. The estimated coefficients of the interaction of ISPO implementation and large firms are negative and statistically significant across all specifications. These results confirm that the ISPO policy significantly reduced the performance of large palm oil firms. This evidence is supported by the negative sign of the variable Post ISPO policy suggesting that palm oil firms performed poorer in 2015, compared with 2010. Yet, the positive and significant estimated coefficients on large firms across the four models imply that large firms outperform their counterparts (medium).

The coefficient of the interaction term in Model 1 reveals that ISPO policy reduced firms' performance by IDR 75 m (USD 5250) when controlling for other variables in the RD estimation. The result confirms the precise estimate of RD estimation. The reduction in performance of this sample of firms implies that ISPO significantly reduced the performance of the treated firms. Furthermore, the adjusted-$R^2$ for the model is 0.46, which is the highest among the four specifications. This indicates the goodness of fit of RD analysis on ISPO policy evaluation. Meanwhile, Model 3 showed that an increase in the number of employees reduced large firms' performance by IDR 860,000, while ISPO was put into effect. These findings suggest that the reduction in performance induced by ISPO increased with the scale of large companies. Yet, interestingly, the fourth model showed that the performance of firms located in provinces that implemented the REDD+ policy are better off than firms in provinces that did not implement the policy. The estimated coefficient on the interaction of Post_ISPO and REDD+ is IDR 135 m, which suggests that the REDD+ policy presented large firms with the opportunity to absorb sustainability measures ahead of the introduction ISPO policy.

Concerning other selected variables, we revealed that the adoption of technologies for treating palm oil mill effluent significantly reduces firms' performance. This highlights the lack of capital and skills to invest in efficient facilities. Meanwhile, Model 3 showed that firms located in the industrial zone are better-off than firms located outside the industrial zone. In addition, the results show that domestic capital significantly increases firms' performance. Contrariwise, foreign capital significantly reduced firms' performance. This is in part possibly related to the Indonesian foreign investment laws and policies. Furthermore, model specifications showed that the scale of rainfall at the provincial level reduces the performance of firms. This is because Indonesia experienced a devastating drought in 2015 in which at least 11 provinces [46] were hit severely. In contrast, the models showed that temperatures and water supply at the provincial level significantly increased firms' performance. These results are in accordance with previous studies, which show that water and temperature are two important environmental factors affecting the supply of palm oil.

Land remains an imperative resource for palm oil production; thus, companies may promptly purchase land before the ISPO implementation. In addition, the ISPO policy is aimed at improving the eco-friendly image of the Indonesian palm oil export companies globally. To analyse the ISPO effect on firms with different attributes, we conducted heterogeneity analyses based on land expansion and export status of palm oil firms. The results in Table 6 show a significant reduction in firms' performance post-ISPO among firms that purchased land in 2010. The estimated coefficient of the interaction term of Post_ISPO and land expansion is IDR −47,000, whereas the coefficient of land expansion is insignificant on large firms. This indicates that land expansion compromises large firm's economies of scale. However, results in the interaction term of post ISPO and export suggest that ISPO policy does not affect exporting firms.

**Table 5.** Estimation results of the effect of ISPO policy on firm's performance using DID.

| Variables | 1 Coef. | | S.E | 2 Coef. | | S.E | 3 Coef. | | S.E | 4 Coef. | | S.E |
|---|---|---|---|---|---|---|---|---|---|---|---|---|
| Post_ISPO | −119.03 | *** | 16.00 | −166.64 | *** | 34.60 | −89.08 | * | 41.33 | −253.47 | *** | 40.04 |
| Large firm | 14.46 | | 14.60 | 121.02 | ** | 40.50 | 74.98 | *** | 14.92 | 129.81 | * | 46.89 |
| Post_ISPO * Large firm | −74.79 | ** | 21.20 | −234.67 | ** | 69.78 | −112.87 | ** | 27.08 | −250.09 | ** | 84.92 |
| N. of employees | | | | | | | 0.37 | ** | 0.09 | | | |
| Post_ISPO * N. of employees | | | | | | | −0.86 | ** | 0.23 | | | |
| REDD+ | | | | | | | | | | −113.32 | * | 42.22 |
| Post_ISPO * REDD+ | | | | | | | | | | 135.05 | * | 54.79 |
| Domestic capital | 0.03 | ** | 0.01 | 0.03 | | 0.05 | −0.10 | | 0.09 | 0.03 | | 0.04 |
| Foreign capital | −0.28 | *** | 0.03 | 0.14 | | 0.26 | 0.09 | | 0.17 | 0.17 | | 0.25 |
| Industrial zone | 5.33 | | 2.69 | 17.91 | | 18.51 | 36.01 | | 22.34 | 14.18 | | 16.79 |
| Technology | 0.05 | | 2.19 | −3.72 | | 2.41 | −4.19 | ** | 1.15 | −3.48 | | 2.30 |
| Energy intensity | −3162.11 | | 1770.72 | −46.56 | | 661.26 | 54.71 | | 511.44 | −82.24 | | 585.26 |
| Population | 0.00 | | 0.00 | −0.01 | | 0.01 | 0.00 | | 0.00 | −0.01 | | 0.01 |
| Rainfall | −1.13 | | 0.10 | −4.28 | ** | 1.17 | −3.74 | *** | 0.48 | −3.52 | | 1.24 |
| Temperature | 2.13 | | 0.33 | 0.82 | | 1.60 | 0.74 | | 1.28 | 3.57 | ** | 2.22 |
| Education | 0.31 | | 0.07 | 0.05 | | 0.14 | 0.04 | | 0.08 | −0.02 | | 0.08 |
| Water | −0.14 | | 0.05 | 0.22 | ** | 0.09 | 0.14 | ** | 0.04 | 0.25 | | 0.28 |
| Agricultural land | 0.45 | | 11.23 | −65.34 | | 70.23 | −41.16 | | 43.54 | −29.69 | | 32.00 |
| Papua | | | | 35.19 | | 35.19 | 23.07 | | 186.50 | −74.11 | | 217.80 |
| Sulawesi | 28.30 | | 13.83 | −29.95 | | −29.95 | −25.84 | | 78.61 | −99.69 | | 90.44 |
| Kalimantan | 67.31 | | 32.60 | 5.36 | | 5.36 | 5.74 | | 125.02 | −90.35 | | 146.41 |
| Sumatera | 43.19 | | 21.68 | −14.58 | | −14.58 | 3.34 | | 89.85 | −76.42 | | 125.66 |
| Constant | −62.31 | | 31.19 | 165.56 | | 174.11 | 79.03 | | 63.71 | 203.14 | | 135.31 |
| Adj-$R^2$ | 0.46 | | | 0.16 | | | 0.25 | | | 0.17 | | |
| N | 158 | | | 406 | | | 406 | | | 406 | | |

Note: *** $p < 0.01$, ** $p < 0.05$, * $p < 0.1$. S.E is adjusted with cluster.

**Table 6.** Results of heterogeneity analysis of ISPO policy by firm type.

| Type of firms | (1) | | | (2) | | |
|---|---|---|---|---|---|---|
| | Coef. | | S.E | Coef. | | S.E |
| Post_ISPO | −134.57 | ** | 32.23 | −225.81 | *** | 28.61 |
| Large firm | 88.65 | | 45.43 | 135.71 | ** | 48.66 |
| Post_ISPO * Large firm | −201.17 | * | 80.66 | −263.73 | ** | 86.97 |
| Land expansion in 2010 | 0.04 | *** | 0.00 | - | - | - |
| Post_ISPO * Land expansion in 2010 | −0.05 | *** | 0.00 | - | - | - |
| Export | - | - | - | −27.62 | - | 63.34 |
| Post_ISPO * Export | - | - | - | −145.62 | - | 106.51 |
| Other control variables | | Yes | | | Yes | |
| **Adj-R²** | | 0.27 | | | 0.17 | |
| **N** | | 406 | | | 406 | |

*** $p < 0.01$, ** $p < 0.05$, * $p < 0.1$; S.E is adjusted with cluster.

*4.3. Robustness Check*

Lastly, we conduct robustness checks for our results. Given the choice of assignment variable, which might lead to different policy effects, we introduced falsification tests into the RD model. First, we applied the Malaysian definition of large firms, which is 150 employees, as the cut-off point. Next, we applied Thailand's definition of large firms, which is 200 employees. We used this specification for the reason that both countries are major producers of palm oil, with Malaysia ranked second in the world and shared the border with Indonesia in Kalimantan Island.

The first falsification test in Table 7 shows a significant false treatment effect for large palm oil firms around the introduced cutoff under the standard RD model, whereas there is no significant effect when covariates are included. Meanwhile, Test 2 exhibits statistically insignificant results. The sign of the false treatment effect is negative in both scenarios. Accordingly, these aforementioned falsification tests indicate that the RD approach captures a valid impact of ISPO policy on firm performance around the cut-off.

**Table 7.** Robustness check for the treatment effect of ISPO policy on palm oil firms' performance.

| Performance | Test 1 | Test 2 |
|---|---|---|
| Standard | −63.07 * | 49.23 |
| | (38.2) | (43.56) |
| Conventional | −12.79 | −11.04 |
| | (−33.5) | (41.87) |
| N | | 406 |

Standard error is presented in parentheses. * $p < 0.1$.

## 5. Conclusions

By identifying the impact of ISPO policy on palm oil firms' performance, we revealed that the mandatory and punitive sustainable production policy negatively and significantly reduced the financial performance of firms. The negative effect of the policy increased with firm size with the exception of firms that experienced corresponding international sustainable production measures and adjusted themselves to these standards. Furthermore, our analysis showed that the land expansion before the introduction of the policy compromises large firms' performance. Overall, our analysis suggests that the introduction of the policy imposed extra pressure on palm oil firms and subsequently positioning large firms in a disadvantaged position. This is because the policy may have forced firms to invest in technologies that reduce greenhouse emissions, increase energy efficiency, and reduce waste.

The empirical findings of this research have important implications for policymakers to promote sustainable palm oil production while achieving economic benefits across the sector. First, the positive effect of ISPO on firms that adopted early international standards supports a contingency view of the need for conducting pilot projects to prepare firms for ISPO certification. Therefore, policymakers should conduct pilot projects in readiness for ISPO implementation. Second, financial loss supports the view of insufficient resources for adopting the standards of the policy. Thus, the government should provide capacity building and funding in order for the policy or standards to materialise. In particular, rather than imposing punishment, which is revoking companies' licences for failure to comply with the sustainability policy. The government should provide incentives to encourage companies to adhere to the policy. Furthermore, since the management of water, energy, fossil fuel, and waste have to be in accordance with ISPO standards. The government should provide these firms with alternative techniques or cushion these firms towards these changes by providing training or other forms of knowledge exchange. Thus, offering cost-effective ways for these firms to invest in mechanised and technological advance production facilities that meet the goal of the policy.

Finally, while the study theoretically highlights the impact of a sustainable production policy on palm oil firm performance. Some limitations and future research lines are noted. The study focuses on two periods. However, natural disasters such as droughts and global shocks could have an indirect or direct impact on business performance even though their impacts are likely to be smaller than the policy impacts. Clearly, separating and comparing natural disaster shocks with policy impacts would be an interesting topic for future studies. Therefore, there is a need for additional empirical research that broadens the time horizon to capture the dynamics and improve the understanding of the causal effect of the policy. However, these are beyond our scope and limits.

**Author Contributions:** Conceptualization, Y.C.Y.C.; Data curation, N.K.; Investigation, Y.C.Y.C.; Methodology, Y.C.Y.C.; Project administration, Y.C.Y.C.; Software, N.K.; Writing—original draft, N.K. All authors have read and agreed to the published version of the manuscript.

**Funding:** Supported by the Ministry of Science and Technology, Taiwan (R.O.C) [grant number: MOST 109-2621-M-020-001–MY2].

**Conflicts of Interest:** The authors declare no conflict of interests.

**Appendix A**

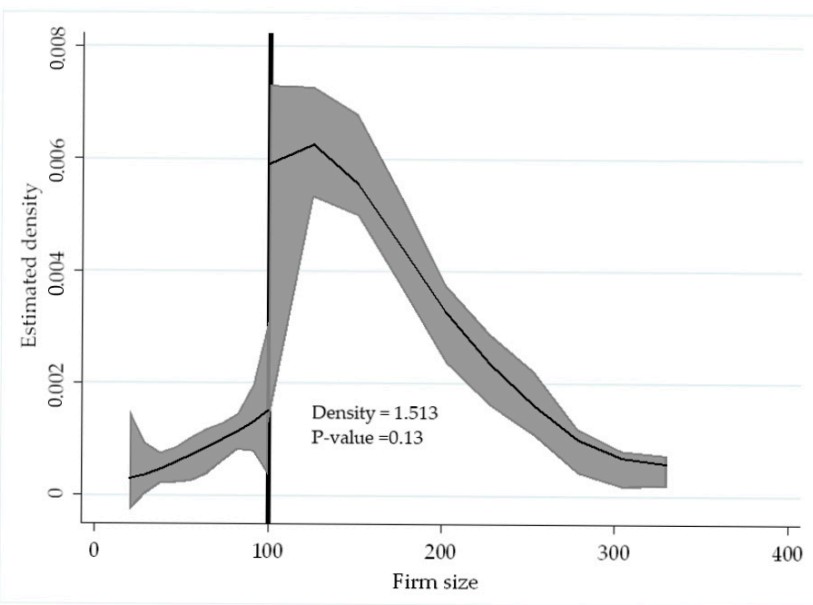

**Figure A1.** Density of running variable.

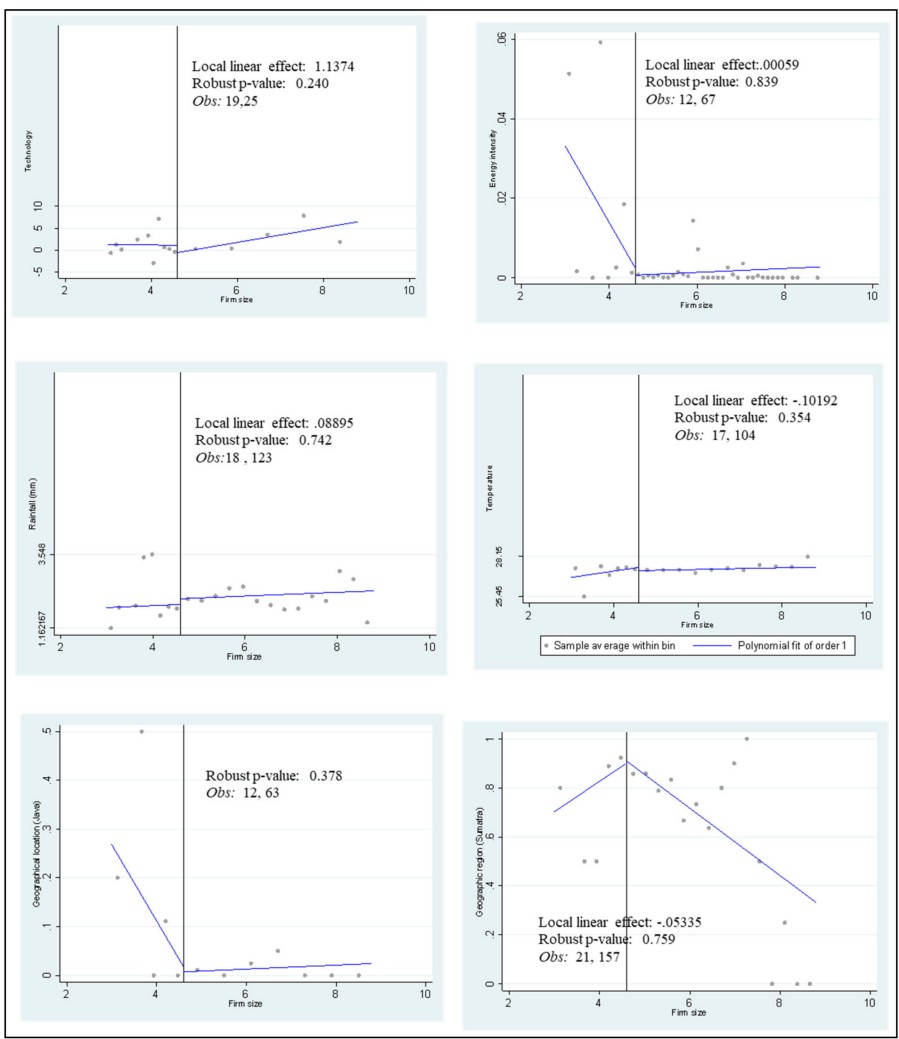

**Figure A2.** Covariates balance. This figure presents RD effects on predetermined covariates, namely technology, energy-intensity, rainfall, temperature, and geographical location (Java and Sumatra), respectively.

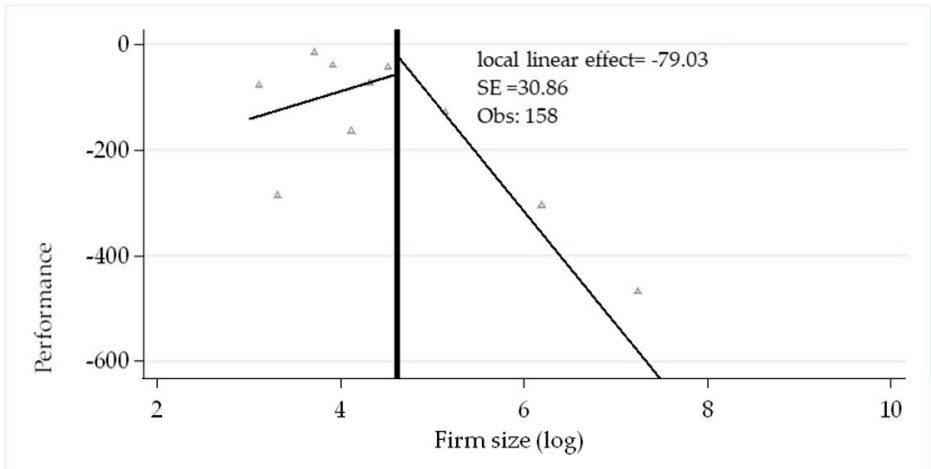

**Figure A3.** Effect of ISPO policy on firm performance.

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
