# Peer review of "Sustainable Production Policy Impact on Palm Oil Firms’ Performance: Empirical Analysis from Indonesia"

_sustainability, doi:10.3390/su12208750_

Round 1

Reviewer 1 Report

-The topic of the paper is under high relevance and interest. However, the relevance of the research is not revealed from scientific point of you.

-There is a lack of scientific discussion in the introduction part. The concept and interpretation of the firm performance is not revealed. It doesn't clear how authors interpret it in the current study.

-I recommend to review the presentation of the results in fig. 1 and fig. 2, the information seems not really important, representing the results.

Author Response

Please see the attached response letter for Reviewer 1

Reviewer 2 Report

Overall, I cannot understand how the authors have reached the conclusion that the introduction of ISPO was the cause of the decline in fortunes of large palm oil producers between 2010 and 2015. All producers were generating profits in 2010, with the large producers generating larger profits than the medium producers. and large). In 2015, they were all generating losses, with large producers generating larger losses than the medium producers. This seems as would be as expected, with large producers spending more and hence losing more. Figure 2 appears to show two data sets with exponential relationships in each - you've just chosen the draw a boundary in each and plot a straight trend line on either side of the boundary. If you removed your artificial boundary lines and just drew exponential graphs on each they would look very similar (just inverted due to the fact that 2015 was a loss-making financial environment). Figure 2b also has very few data points so the conclusion about there being a discontinuity seems weak. 

How can you be sure that the introduction of ISPO made these losses bigger than they would otherwise have been? What was causing the losses in 2015 for all firms? Was it global palm oil prices? Something else? Were these factors just greater for large firms due to their greater exposure to these risks? Presumably the introduction of ISPO would have made certain costs greater (e.g. land, labour, compliance costs) - what were these?

Other comments:

Grammar needs improving throughout the article. For example, the following few sentences in the introduction contain five errors (indicated in [] after each one). A full proof read of the whole article is required.

"In particular, government[s] act in response by using the notion of domestic governance so as to gain sovereignty. One notable example, [no comma needed] is the formulation of the Indonesian Sustainable Palm Oil policy (ISPO[)] by the Indonesian government, [no comma needed] as a counterpart to the private and globally accepted Roundtable on Sustainable Palm Oil (RSPO) standard. Nevertheless the emergence of these new governance structures changes the nature through which companies operates [no s required]."

Table 1 - Papau should be Papua

Table 2 - Unclear what figures beneath average profit are - e.g. (473.1) (395) etc. Also should say "Mean change between 2010 and 2015" (2015 is cut off)

Author Response

Please see the attached response letter for Reviewer 2

Reviewer 3 Report

Sustainable production of palm oil, especially in Indonesia is very significant issue in the world to achieve the SDG 12. And the reviewer believes the authors' work seems to be original in terms of using the Indonesian statistics. However, some sentences remain confusing for readers, so please revise in accordance with following suggestions.

line 40; The word "carry" should be changed to  "carried",

line 51-; The authors must indicate a reference of "taking the center stage",

line 64-; Did the government introduce ISPO firstly in 2011 or 2015? Please keep consistency in your manuscript (ex. line 164),

line 250 etc.; Is the word "casual" really correct? Please check all in the manuscript (ex. line 251, 257,,,),

line 324-326 & 330-333; The sentences seem to be very similar, please think about integrating these two sentences,

line 335; Table 2 is insufficient, in term of meaning of the last line and between 2010 and when?

Overall; Did the international price change of palm oil affect to each company's performance during 2010 to 2015? If possible, please add some explanations.

Author Response

Please see the attached response letter for Reviewer 3

Round 2

Reviewer 2 Report

Thank you for taking the time to revise the manuscript and respond to my comments. Your response has clarified some of the results and the assumptions employed in your analysis. However, I feel that this updated information has only reinforced my previous concerns that the results do not justify the conclusion that the introduction of ISPOs had a major effect on the financial performance of palm oil firms in Indonesia.

There does not appear to be a major difference between large and medium firms with regards to their declining financial performance between 2010 and 2015. One group declined by 230% and the other by 222%. In your response you first say large firms declined by 230% and medium firms declined by 222%. Later you say that this was the other way around (i.e. medium firms actually declined more in percentage terms). However, whichever way around this should be, the numbers are very similar. There was obviously something else going on in 2015 that was much more significant than the introduction of ISPOs, which you speculate may be drought and low palm oil prices. It is inevitable that these factors will have affected large and medium firms differently. Given how important these other factors must have been in 2015, I can’t see how you can conclude that “The policy imposed an average treatment effect of a decline in profit of IDR 234.14m on the treated group (large firms)”. IDR 234.14m is simply the difference in average performance between large and medium firms due to ALL factors and cannot be attributed entirely to the ISPO policy.

The OECD report about small-medium firms being more affected by the global financial crisis doesn’t seem all that relevant. That impact was in 2007-2008 and was due to very different causes. I’m not sure how relevant it is for understanding the impacts of drought and low palm oil prices in 2015.

The selection of large firms as a treatment group and medium firms as a control group seems to me to be a major methodological flaw. Perhaps I don’t understand the RD method fully, but if you are going to describe them as “treatment” and “control” groups then they should be identical in every way apart from the factor being investigated (i.e. the role of ISPOs). In this case, the two groups have major differences apart from the imposition of ISPOs because they are different sizes. Large firms will inevitably have different cost structures and relationships that will lead to different impacts on financial performance at a time of low palm oil prices. There may also be geographical differences that will lead to differing impacts of drought (e.g. maybe large firms were concentrated in areas that were more affected by drought in 2015). The DID analysis appears to consider some of these factors by controlling the effect of firm characteristics, province characteristics and location factors, but I really can’t understand how it does this. The methodology provides very little detail on how these factors are taken into account and the numbers in Table 3 don’t make sense (e.g. how can medium firms have an average of 175 employees when the definition for a medium firm is a firm with between 20 and 99 employees?)

Author Response

Dear Reviewer,

Thank you for your comments.

Revised manuscript and response letter are attached.
